# Autoimmune Regulator Gene Polymorphisms and the Risk of Primary Immune Thrombocytopenic Purpura: A Case-Control Study

**DOI:** 10.3390/ijms24055007

**Published:** 2023-03-05

**Authors:** Muhammad T. Abdel Ghafar, Ola A. Elshora, Alzahraa A. Allam, Raghda Gabr Mashaal, Shereen Awny Abdelsalam Hamous, Sarah Ragab Abd El-Khalik, Rania Nagi Abd-Ellatif, Reham A. Mariah, Radwa Eissa, Mai Mwafy, Rasha E. Shalaby, Elham Nasif, Rasha A. Elkholy

**Affiliations:** 1Department of Clinical Pathology, Faculty of Medicine, Tanta University, Tanta 31511, Egypt; 2Department of Internal Medicine, Faculty of Medicine, Tanta University, Tanta 31511, Egypt; 3Department of Medical Biochemistry, Faculty of Medicine, Tanta University, Tanta 31511, Egypt; 4Department of Microbiology and Immunology, Faculty of Medicine, Tanta University, Tanta 31511, Egypt; 5Department of Basic Medical Sciences, Faculty of Medicine, Galala University, Galala 43511, Egypt; 6Department of Physiology, Faculty of Medicine, Tanta University, Tanta 31511, Egypt

**Keywords:** immune thrombocytopenic purpura, autoimmune regulator gene, polymorphism, haplotypes, risk

## Abstract

This study aimed to assess the possible association between two single nucleotide polymorphisms (SNPs) of the autoimmune regulator (*AIRE*) gene (rs2075876 G/A and rs760426 A/G) with the risk of primary immune thrombocytopenia (ITP), as well as AIRE serum levels, in the Egyptian population. In this case-control study, 96 cases with primary ITP and 100 healthy subjects were included. Two SNPs of the *AIRE* gene (rs2075876 G/A and rs760426 A/G) were genotyped via Taqman allele discrimination real-time polymerase chain reaction (PCR). Additionally, serum AIRE levels were measured using the enzyme-linked immunosorbent assay (ELISA) technique. After adjusting for age, gender, and family history of ITP, the *AIRE* rs2075876 AA genotype and A allele were associated with increased ITP risk (adjusted odds ratio (aOR): 4.299, *p* = 0.008; aOR: 1.847, *p* = 0.004, respectively). Furthermore, there was no significant association between *AIRE* rs760426 A/G different genetic models and ITP risk. A linkage disequilibrium revealed that A-A haplotypes were associated with an increased ITP risk (aOR: 1.821, *p* = 0.020). Serum AIRE levels were found to be significantly lower in the ITP group, positively correlated with platelet counts, and were even lower in the *AIRE* rs2075876 AA genotype and A allele, as well as A-G and A-A haplotype carriers (all *p* < 0.001). The *AIRE* rs2075876 genetic variants (AA genotype and A allele) and A-A haplotype are associated with an increased ITP risk in the Egyptian population and lower serum AIRE levels, whereas the SNP rs760426 A/G is not.

## 1. Introduction

Immune thrombocytopenia (ITP) is an autoimmune bleeding disorder caused by excessive immune-mediated platelet destruction and inadequate bone marrow production [1]. However, the pathogenesis of ITP is relatively complicated, and the precise etiology and pathogenesis remain unknown. Several environmental factors, including viral infection, autoimmune disorders, and medications, have been linked to ITP pathogenesis. In addition, genetic factors have been identified as a risk factor for the development of ITP [2].

Recent evidence suggests that several mechanisms involving regulatory B- and T-cells, as well as natural killer cells, myeloid-derived suppressor cells, and dendritic cells [3,4,5,6] are all responsible for thrombocytopenia that develops in patients with ITP [7]. However, the mechanism underlying autoreactive T- and B-cell generation in autoimmune diseases remains unclear.

Autoimmune Regulator (*AIRE*) gene is one of the key genes that regulate immune tolerance, making it a candidate for better understanding the pathogenesis of autoimmune disorders [8]. It is located on chromosome 21q22.3 region and has an 11.9 kb genomic region with 14 exons. It encodes a 545 amino acid transcriptional regulator protein of 58 kD. This protein promotes the negative selection of thymic autoreactive T-cells by controlling the expression of a diverse set of self-antigens known as tissue-restricted antigens. The AIRE protein could promote the proteasome pathway, enhancing cell trafficking and DNA repair, as well as self-tolerance via its E3 ubiquitin ligase activity [9,10]. In addition to medullary thymic epithelial cells, evidence suggests that AIRE is also expressed in peripheral lymphoid organs [11]. Extra-thymic *AIRE*-expressing cells were discovered as a distinct “bone-marrow-derived antigen-presenting CD^45low^” population that functionally inhibits effector (CD4^+^) T-cells, preventing co-stimulation and inducing tolerance [12].

Several single nucleotide polymorphisms (SNPs) in the *AIRE* gene have been identified and reported to be associated with several autoimmune disorders, such as rheumatoid arthritis (RA) [13,14,15], systemic lupus erythematosus (SLE) [16,17], autoimmune thyroiditis, and systemic sclerosis [18], as well as myasthenia gravis [19], vitiligo [20], type 1 diabetes mellitus [21], and autoimmune hepatitis [22]. However, only two SNPs (rs2075876 and rs760426) have received attention due to their significant association with autoimmune disorders, particularly RA and SLE [14], as well as their impact on AIRE expression, particularly the A allele of rs2075876.

Experimental studies have shown that *AIRE* SNPs alter *AIRE* gene transcription, altering peripheral tissue antigen expression that controls peripheral antigen presentation, providing less efficient negative selection, promoting autoimmune T-cell survival, and increasing susceptibility to autoimmune diseases [23]. Furthermore, depending on the strain’s genetic background, *AIRE*-deficient mice can exhibit a wide range of autoimmune phenomena [24], including multi-organ lymphocytic infiltration and circulatory antibodies [25], implying a critical role in setting a self-tolerance threshold in immune regulation [26].

On the basis of these data, the *AIRE* gene may be a candidate gene for several autoimmune disorders. However, the association between *AIRE* gene polymorphisms and ITP risk has yet to be investigated. Therefore, our study aimed to assess the possible association between two SNPs (rs2075876 G/A and rs760426 A/G) of the *AIRE* gene with the risk of primary ITP, as well as AIRE serum levels, in the Egyptian population.

## 2. Results

### 2.1. Basic Characteristics of the Studied Cohorts

In this study, 96 patients with primary ITP and 100 healthy subjects were included. They did not differ significantly in terms of age or gender. However, the ITP group had a significantly higher proportion of positive family history and a lower platelet count than the healthy control group (Table 1)**.**

### 2.2. AIRE SNPs and ITP Risk

The Hardy–Weinberg equilibrium (HWE) of different genotypes of the two *AIRE* SNPs (rs2075876 G/A and rs760426 A/G) in the control group was confirmed, indicating that our sample size was sufficient to adequately represent our population. The *AIRE* rs2075876 AA genotype and A allele were more frequent in the ITP group than in the control group (*p* = 0.004 and 0.005, respectively; Table 2). After adjusting for age, gender, and family history of ITP, the ITP risk was found to be higher in the *AIRE* rs2075876 AA genotype (adjusted odds ratio (aOR): 4.299, 95% confidence interval (CI): 1.650–11.202, *p* = 0.008) and A allele carriers (aOR: 1.847, 95% CI: 1.209–2.822, *p* = 0.004) than in the GG genotype and G allele carriers. Additionally, an increased ITP risk was observed under the recessive genetic model (aOR: 3.257, 95% CI: 1.406–7.545, *p* = 0.005) after adjusting to the confounders, whereas the dominant model revealed no association (Table 3). Furthermore, there was no significant difference in the distribution of *AIRE* rs760426 A/G different genetic models between ITP and healthy control groups.

### 2.3. AIRE SNP Haplotypes and ITP Risk

A linkage disequilibrium was detected between the two *AIRE* SNPs studied (D′ = 0.89), revealing four main haplotypes: G-A; G-G; A-G; and A-A. After adjusting for the confounders, A-A haplotypes were more frequent in the ITP group than in the control group and were associated with an increased ITP risk (aOR: 1.821, 95% CI: 1.099–3.017, *p* = 0.020) (Table 4).

### 2.4. AIRE SNPs and Haplotypes and Serum AIRE Levels

Serum AIRE levels were found to be significantly lower in the ITP group than in the control group (median, 3.1 ng/mL (IQR, 3.2) vs. median, 5.5 ng/mL (IQR, 3.0), respectively, *p* < 0.001; Figure 1A). Furthermore, a positive correlation was found between serum AIRE levels and platelet counts (*r* = 0.535, *p* < 0.001; Figure 1B). Serum AIRE levels differed significantly among different *AIRE* rs2075876 genotype and allele carriers (all *p* < 0.001), with the lowest levels detected in the AA genotype (Figure 1C) and A allele (Figure 1D) carriers (all *p* < 0.001). In contrast, AIRE serum levels did not differ between different *AIRE* rs760426 genotypes (*p* = 0.438; Figure 1E) and alleles (*p* = 0.748; Figure 1F). Furthermore, serum AIRE levels were significantly lower in the A-G and A-A haplotypes than in the G-A and G-G haplotypes (Figure 1G).

## 3. Discussion

The hallmark of the normal adaptive immune system is the induction of self-tolerance. For instance, the *CD40* gene SNP (rs1883832 C>T) is associated with an increased risk of ITP, particularly when combined with *CD40* rs4810485 G>T in the Egyptian population [27,28]. In addition, the *histone deacetylase 3* gene SNP (rs2530223 C>T) was found to be associated with increased susceptibility to ITP in all genetic models, with TC/TT genotypes associated with severe thrombocytopenia in the Han population [29].

In this regard, AIRE is regarded as a key player in self-tolerance machinery. Variants in the *AIRE* gene have been shown to suppress its transcription and protein levels, reducing negative selection and enhancing autoimmune T-cell survival. Autoimmune T-cells play a central role in promoting the development of a variety of immunological disorders via producing a wide range of autoantibodies [23]. Therefore, it is not surprising that the existence of *AIRE* genetic variations has been linked to a variety of autoimmune disorders [30].

Notably, only two of the eleven SNPs identified in the *AIRE* gene (rs2075876 G/A and rs760426 A/G) have received attention and have been found to carry a risk for several autoimmune disorders, particularly RA and SLE [17,31]. Although rs2075876 G/A and rs760426 A/G exist in non-coding intronic regions, they have been implicated in inhibiting *AIRE* gene expression, thereby impairing thymic negative selection and increasing the risk for autoimmune disorders [32,33,34]. Although the mechanisms underlying RA and SLE are unknown, evidence suggests that the loss of self-tolerance with the activation of autoreactive T- and B-cells plays a role [35,36]. In the same vein, ITP is characterized by the excessive activation and proliferation of platelet autoantigen reactive cytotoxic T lymphocytes [37], abnormal T helper cells [38], and B-cells [39], indicating a shared pathogenicity with RA and SLE in terms of altered immune tolerance. Therefore, we conducted this study to assess, for the first time, the association between these two *AIRE* SNPs and ITP risk in the Egyptian population.

Our study revealed that *AIRE* rs2075876 G/A was associated with increased ITP risk under the additive, recessive, and multiplicative genetic models, with the AA genotype and A allele conferring the higher ITP risk. Other relevant studies revealed that *AIRE* rs2075876 plays a crucial role in determining the risk for various autoimmune disorders, such as RA and SLE, which is consistent with our findings. In the Chinese population, for instance, the *AIRE* rs2075876 A allele has been reported to increase RA risk under recessive [31,40] as well as dominant and co-dominant genetic models [13,40]. Furthermore, A. Moneim et al. reported that the *AIRE* rs2075876 A allele was more frequent in patients with RA than in the healthy subjects under both co-dominant and over-dominant models [15]. Recently, Salesi et al. reported that *AIRE* rs2075876 homozygous AA and heterozygous AG genotypes increased RA risk when compared to the GG genotype in the Iranian population [41]. In addition, Attia et al. reported an association between this polymorphism and SLE risk in the Egyptian population [17]. On the contrary, Alghamdi et al. found that the *AIRE* rs2075876 A allele was associated with a low risk of SLE in the Egyptian population [16]. However, no association between this polymorphism and the SLE risk has been reported in the Mexican population [42].

On the other hand, our study findings did not find any significant association between the different genetic models of *AIRE* rs760426 and ITP risk in our population. In contrast, Shao et al. and Li et al. found that the *AIRE* rs760426 G allele was more frequent in patients with RA than in healthy controls and was associated with higher RA risk under the recessive model in the Chinese population [31,40]. However, a borderline association between rs760426 and RA risk was detected by Feng et al. [13]. Moreover, A. Moneim et al. found that the *AIRE* rs760426 AG genotype and G allele were more frequent in patients with RA than in healthy controls in both co-dominant and over-dominant genetic models [15]. Furthermore, Attia et al. reported that the *AIRE* rs760426 homozygous genotype was more frequent in patients with SLE than in healthy controls, indicating a stronger association with SLE risk in the Egyptian population [17].

Since the association between the two studied polymorphisms and autoimmune disorders has been reported, such as in a previous meta-analysis that revealed an association between these two SNPs and high RA risk under all genetic models [23], it is reasonable to assume that linkage disequilibrium may exist between these polymorphisms and may confer a risk for autoimmune disorders. In this study, such linkage disequilibrium was observed, with the A-A haplotype being associated with a 1.821-fold increased ITP risk. In contrast, A. Moneim et al. revealed no evidence of linkage disequilibrium between the two SNPs in RA [15]. In terms of other *AIRE* SNPs, several SNPs were examined in combination and revealed that *AIRE* haplotype CCTGCC (AIRE C-103T, C4144G, T5238C, G6528A, T7215C, and T11787C) showed a three-fold increased risk for vitiligo [20]. Furthermore, the *AIRE* CCTGCT and CGTGCC haplotypes (C-103T, C4144G, T5238C, G6528A, T7215C, and T11787C) showed a 9.47- and 3.51-fold increased risk for alopecia areata, respectively [43]. Furthermore, the haplotype of five *AIRE* SNPs (rs2075876, rs2075877, rs933150, rs1003854, and rs1078480) and the two SNPs (rs2256817 and rs760426) revealed a 1.852- and 1.950-fold increased risk for RA, respectively [13]. On the other hand, the variant effect prediction for the two studied SNPs in the study revealed that 36% of them are upstream gene variants, and 36% are intronic variants. In contrast, other *AIRE* SNPs (rs2075876, rs2075877, rs933150, rs1003854, and rs1078480) were 43% intronic variants and 25% noncoding transcript variants.

In this study, we estimated serum AIRE levels in the studied groups to investigate the potential impact of *AIRE* genetic variants (rs2075876 G/A and rs760426 A/G) on its expression levels. Our findings revealed that the *AIRE* rs2075876 AA genotype and A allele had a strong impact on AIRE expression, lowering its serum levels. This is the first study to investigate the relationship between *AIRE* SNPs and their protein expression levels in autoimmune disorders; no previous studies have investigated this issue so far. However, this effect was demonstrated in silico using data from the Gene Expression Omnibus (GEO) database for 210 lymphoblastoid cells, revealing that the AIRE rs2075876 A allele was significantly correlated with decreased *AIRE* transcription and, thus, decreased protein levels [14]. However, GEO revealed no association between the *AIRE* rs760426 G allele and *AIRE* expression [44].

Although this study was the first to investigate the association between *AIRE* rs2075876 G/A and rs760426 A/G genetic variants and haplotypes with ITP risk, as well as serum AIRE levels, it does have some limitations that should be considered. First, this is a single-center study. Second, because this study was conducted on a single population, our findings cannot be generalized to other populations. Third, no functional analysis was conducted to assess the causal relationship between *AIRE* SNPs on one side and *AIRE* expression and ITP pathogenesis on the other. Fourth, because the samples were collected prior to any treatment, the impact of treatment, particularly corticosteroids, on serum AIRE levels could not be investigated. Finally, serum AIRE levels were not correlated with the duration of ITP. Further studies on other populations and *AIRE* SNPs, as well as the impact of ITP treatment on AIRE levels and the relationship between AIRE levels and ITP duration, are warranted to support our findings.

## 4. Materials and Methods

### 4.1. Study Cohort

In this case-control study, 96 patients with primary ITP at their initial presentation were recruited from the hematology unit of the Internal Medicine Department, Tanta University Hospitals, as a case group. Their inclusion criteria were all cases diagnosed with ITP based on the standard criteria of the International Working Group of isolated thrombocytopenia (platelet counts of less than 100 × 10^3^/µL in the absence of other causes of thrombocytopenia) [45]. Patients with secondary causes of ITP such as viral infections, drugs, and autoimmune disorders, such as RA, SLE, and Chron’s disease, were excluded from the study.

In addition, 100 healthy subjects with normal platelet counts were recruited from those attending the out-patient clinics at Tanta University Hospitals as a control group. The exclusion criteria of the case group were also applied to the control group. This study was conducted in accordance with the Helsinki Declaration and was approved by the local ethical committee of the Faculty of Medicine, Tanta University (approval code no. 35353/3/22). Informed written consent was obtained from all the participants.

### 4.2. Clinical and Laboratory Assessment

All participants were asked about their personal and family history of ITP, and patients with ITP were examined clinically and radiologically via ultrasonography to detect splenomegaly. Data on complete blood count (CBC), bone marrow aspiration cytology, antinuclear antibodies (ANAs), and direct anti-globulin test (if Evan syndrome was suspected) used to diagnose ITP, as well as data on viral screening for hepatitis C, cytomegalovirus, human immunodeficiency virus, and helicobacter pylori antigen in stool to exclude secondary ITP, were obtained from the patients’ medical records.

### 4.3. Sampling

Blood samples were collected prior to receiving any treatment. Five milliliters of venous blood was collected via standard venipuncture, and two milliliters was delivered into a K2 EDTA vacutainer tube for genotyping. The remainder was delivered into a gel tube at room temperature, centrifuged at 3000 rpm for 5 min, and the serum was then separated and immediately frozen in −20 °C for the assay of AIRE serum levels.

### 4.4. AIRE SNP Genotyping

The genomic DNA was extracted from blood samples using the GeneJet genomic DNA purification kit (Thermo-Fisher Scientific, Waltham, MA, USA), according to the manufacturer’s instructions and kept at −80 °C for *AIRE* genotyping. Two *AIRE* SNPs (rs2075876 G/A and rs760426 A/G) were genotyped using real-time polymerase chain reaction (PCR) by allelic discrimination assay with Taqman SNP genotyping assay kit (Thermo-Fisher Scientific, MA, USA). In a total volume of 25 µL, the extracted DNA (25 ng/3 µL) was mixed with 2xTaqman genotyping master mix (12.5 µL), 20xTaqman SNP genotyping assay (1.25 µL), and nuclease-free water. The reaction mixture was then run on an Applied Biosystem StepOne real-time PCR instrument (Foster City, CA, USA) under the following thermal cycling conditions: pre-denaturation at 95 °C for 5 min, followed by 40 cycles of denaturation at 95 °C for 15 s and annealing/extension at 60 °C for 60 s. The genotypes were determined with cycle thresholds on the multicomponent plot.

### 4.5. Serum AIRE Assay

Serum AIRE levels were measured using a commercially available enzyme-linked immunosorbent assay (ELISA) kit (Human AIRE, Biovision, Milpitas, CA, USA, catalog no. E4971-100), according to the manufacturer’s instructions, and colorimetrically detected at 450 nm on a Tecan Spectra II Microplate Reader (canton of Zürich, Switzerland). The sample concentration was calculated using a logit–log standard curve. The assay sensitivity was 0.094 ng/mL, with intra-assay and inter-assay coefficients of variation of <8% and <10%, respectively.

### 4.6. Statistical Analysis

All datasets were analyzed using the Statistical Package for the Social Sciences (SSPS) software Version 22 (IBM Corp, Armonk, NY, USA). The data were tested for normality using the Kolmogorov–Smirnov test. The normally distributed numerical variables were presented as mean and standard deviation and were compared using Student’s *t*-test, while the non-normally distributed variables were presented as median and interquartile range (IQR) and were compared using the Mann–Whitney *U* test. The categorical variables were presented as numbers and percentages and were compared using the Chi-square *χ*^2^ test. The significant comparisons were further corrected (Pcorr) using Bonferroni’s correction for multiple testing. The HWE of *AIRE* SNPs (rs2075876 and rs760426) genotypes was determined in the control group using the *χ*^2^ test. The web-based SHEsisPlus platform (http://shesisplus.bio-x.cn/SHEsis.html (accessed on 8 August 2022)) was used for haplotype analysis. The ITP risk was calculated as a crude OR with a 95% CI for both *AIRE* SNP genetic models and haplotypes and then adjusted for confounders such as age, gender, and family history of ITP using multiple logistic regression analysis. A two-sided *p*-value of less than 0.05 was considered statistically significant.

## 5. Conclusions

Our findings reveal that the *AIRE* rs2075876 genetic variants (AA genotype and A allele) and A-A haplotype are associated with increased ITP risk in the Egyptian population and lower serum AIRE levels, whereas the rs760426 A/G SNP is not.

## Figures and Tables

**Figure 1 ijms-24-05007-f001:**
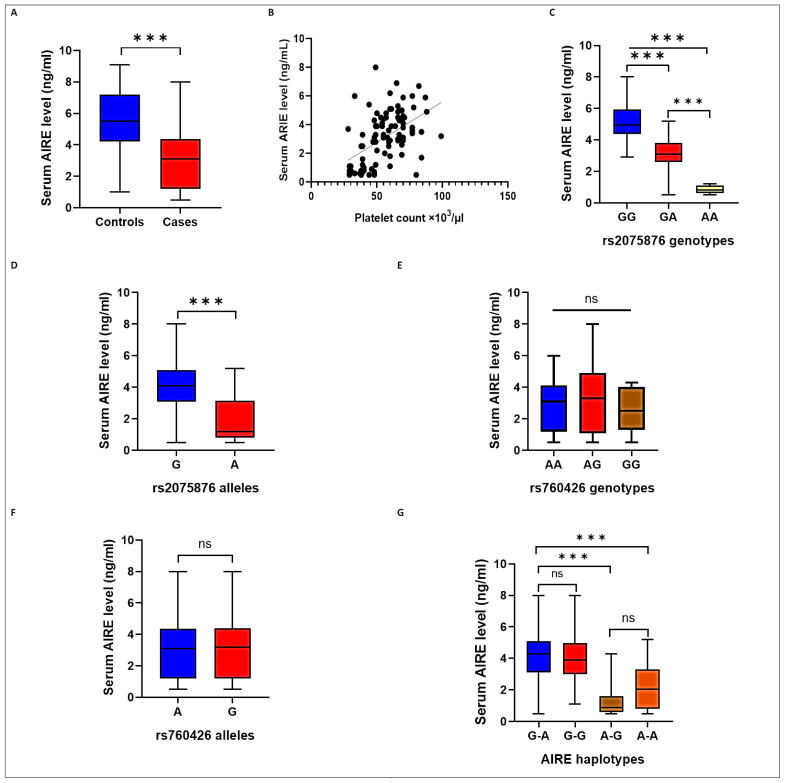
Serum AIRE levels in the studied groups: (**A**) in cases and controls; (**B**) correlated with platelet counts in the ITP group; (**C**) among different *AIRE* rs2075876 genotypes; (**D**) among different *AIRE* rs2075876 alleles; (**E**) among different *AIRE* rs760426 genotypes; (**F**) among different *AIRE* rs760426 alleles; and (**G**) among different *AIRE* haplotypes. *** *p* <0.001, ns, non-significant.

**Table 1 ijms-24-05007-t001:** Basic characteristics of the studied groups.

Characteristics	Case Group (*n* = 96)	Control Group (*n* = 100)	*p*-Value
Age, years, mean ± SD	44.6 ± 12.6	42.3 ± 13.0	0.201
Gender Male, *n* (%) Female, *n* (%)	31 (32.3) 65 (67.7)	40 (40.0) 60 (60.0)	0.262
Positive family history, *n* (%)	23 (24.0)	6 (6.0%)	<0.001 *
Platelets count (×10^3^/μL), median (IQR)	55.5 (23)	210.5 (112)	<0.001 *

*n*—number; SD—standard deviation; and IQR—interquartile range. * *p* < 0.05 vs. control group is significant.

**Table 2 ijms-24-05007-t002:** Distribution of the different genetic models of *AIRE* SNPs (rs2075876 G/A and rs760426 A/G) between the studied groups.

SNPs	Genetic Model		Case Group, *n* (%) (*n* = 96)	Control Group, *n* (%) (*n* = 100)	*χ* ^2^	*p*
*AIRE* (rs2075876 G/A)	Additive	GG	26 (0.27)	41 (41.0)	--	--
		GA	47 (0.49)	49 (49.0)	1.645	0.200
		AA	23 (0.24)	10 (10.0)	8.443	0.004 *
		*HWE p*	--	0.399		
	Dominant	GG	26 (0.27)	41 (0.41)	2.216	0.040 *
		GA/AA	70 (0.73)	59 (0.59)		
	Recessive	GG/GA	73 (0.76)	90 (0.90)	6.815	0.009 *
		AA	23 (0.24)	10 (0.10)		
	Multiplicative (Allele)	G	99 (0.52)	131 (0.655)	7.848	0.005 *
		A	93 (0.48)	69 (0.345)		
*AIRE* (rs760426 A/G)	Co-dominant	AA	43 (0.45)	46 (0.46)	--	--
		AG	47 (0.49)	39 (0.39)	0.703	0.402
		GG	6 (0.06)	15 (0.15)	2.681	0.102
		*HWE p*		0.170		
	Dominant	AA	43 (0.45)	46 (0.46)	0.029	0.865
		AG/GG	53 (0.55)	54 (0.54)		
	Recessive	AA/AG	90 (0.94)	85 (0.85)	3.920	0.054
		GG	6 (0.06)	15 (0.15)		
	Multiplicative (Allele)	A	133 (0.69)	131 (0.655)	0.633	0.426
		G	59 (0.31)	69 (0.345)		

*n*—number; SNP—single nucleotide polymorphism. * *p* < 0.05 is significant.

**Table 3 ijms-24-05007-t003:** Different genetic models of *AIRE* SNPs (rs2075876 G/A and rs760426 A/G) and ITP risk.

SNPs	Model	Genotype	Crude OR (95% CI)	*p*	Adjusted OR (95% CI)	*p*	*p*-_corr_
*AIRE* (rs2075876 G/A)	Additive	GG	Ref	--	Ref	--	
		GA	1.513 (0.803–2.851)	0.201	1.460 (0.754–2.826)	0.262	
		AA	3.627 (1.489–8.835)	0.005 *	4.299 (1.650–11.202)	0.003 *	0.008 *
	Dominant	GG	Ref	--	Ref	--	
		GA/AA	1.871 (1.026–3.413)	0.041 *	1.859 (0.994–3.477)	0.052	
	Recessive	GG/GA	Ref	--	Ref	--	
		AA	2.836 (1.269–6.336)	0.011 *	3.257 (1.406–7.545)	0.006 *	0.005 *
	Multiplicative (Allele)	G	Ref	--	Ref	--	
		A	1.783 (1.188–2.678)	0.005 *	1.847 (1.209–2.822)	0.005 *	0.004 *
*AIRE* (rs760426 A/G)	Co-dominant	AA	Ref	--	Ref	--	
		AG	1.289 (0.712–2.336)	0.402	1.248 (0.675–2.309)	0.480	
		GG	0.428 (0.152–1.203)	0.108	0.421 (0.144–1.233)	0.115	
	Dominant	AA	Ref	--	Ref	--	
		AG/GG	0.952 (0.543–1.672)	0.865	0.985 (0.549–1.767)	0.960	
	Recessive	AA/AG	Ref	--	Ref	--	
		GG	0.378 (0.140–1.019)	0.054	0.380 (0.135–1.064)	0.065	
	Multiplicative (Allele)	A	Ref	--	Ref	--	
		G	0.842 (0.552–1.286)	0.426	0.828 (0.533–1.286)	0.401	

*n*—number; SNP—single nucleotide polymorphism; OR—odds ratio; and CI—confidence interval. * *p* < 0.05 is significant.

**Table 4 ijms-24-05007-t004:** Distribution of different haplotypes of AIRE SNPs between ITP cases and controls.

*AIRE SNP* Haplotypes		Case Group (*n* = 192)	Control Group (*n* = 200)	Crude OR (95% CI)	*p*	Adjusted OR (95% CI)	*p*
G-A	*n* (%)	59 (0.307)	79 (0.395)	---	---	---	---
G-G	*n* (%)	40 (0.208)	52 (0.26)	0.971 (0.570–1.654)	0.913	1.089 (0.623–1.902)	0.765
A–G	*n* (%)	19 (0.099)	17 (0.085)	1.497 (0.717–3.124)	0.283	1.823 (0.836–3.973)	0.131
A–A	*n* (%)	74 (0.385)	52 (0.23)	1.905 (1.168–3.109)	0.010*	1.821 (1.099–3.017)	0.020 *

*n*—number; SNP—single nucleotide polymorphism; and OR—odds ratio; CI—confidence interval. * *p* < 0.05 is significant.

## Data Availability

The data used in this study are available from the corresponding author upon request.

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
