# Peer review of "Autoimmune Regulator Gene Polymorphisms and the Risk of Primary Immune Thrombocytopenic Purpura: A Case-Control Study"

_ijms, 2023, doi:10.3390/ijms24055007_

Round 1

Reviewer 1 Report

In their manuscript „Autoimmune regulator gene polimorphisms and the risk of primary immune thrombocytopenic purpura: A case-control study“ Ghafar et al. report on the impact of gene polymorphisms in the autoimmune receptor gene on the probability of development of ITP in the control group. I enjoyed reviewing this manuscript and can appreciate the amount of work and diligence it took to put this comprehensive dataset together. The topic is of clinical relevance and the results primarily appear to be of importance to further explanation of ITP.In order to ensure the validity of this report, there are some major and minor points that need to be addressed prior to publication. I am fully aware of the complicated nature of a study in single-center setting and the fact that clinical presentation of ITP remains a rather rare disease (0.2 – 0.4/100,000 inhabitants). I still hope that the authors may address the following points, at least to some extent, as this may render their work an excellent and highly citable paper.

Author Response

Dear editor /referee,

Many thanks for your constructive and valuable criticisms. Our responses are presented below and we are looking forward and ready to respond to any future comment (s)

Reviewers' comments:

Reviewer #1:

In their manuscript „Autoimmune regulator gene polimorphisms and the risk of primary immune thrombocytopenic purpura: A case-control study“ Ghafar et al. report on the impact of gene polymorphisms in the autoimmune receptor gene on the probability of development of ITP in the control group. I enjoyed reviewing this manuscript and can appreciate the amount of work and diligence it took to put this comprehensive dataset together. The topic is of clinical relevance and the results primarily appear to be of importance to further explanation of ITP.In order to ensure the validity of this report, there are some major and minor points that need to be addressed prior to publication. I am fully aware of the complicated nature of a study in single-center setting and the fact that clinical presentation of ITP remains a rather rare disease (0.2 – 0.4/100,000 inhabitants). I still hope that the authors may address the following points, at least to some extent, as this may render their work an excellent and highly citable paper.

Response: Thank you very much for recommending our manuscript and your encouraging comments and we are pleased to respond to them.

Major:

  • Primary ITP was defined by the international working group (IWG) as a platelet count less than 100 × 109/L in the absence of other causes or disorders that may be associated with thrombocytopenia. Figure 1 B suggests that also patients containing a platelet count higher than 100 x 109/L but less than 150 x 109/L had been included in this study. Further clarification is needed on the definition by which patients have been included in the case study.

Response: Thank you for your thorough review, and please accept our apologies for our oversight. Actually, our patients were selected based on IWG criteria. Therefore, we decided to revise our data once more and discovered four cases with a platelet count slightly above 100×103/μl. We excluded these patients from our analysis; consequently, the statistical analysis was repeated and the results in the Results section, tables, and figure 1 have been modified after excluding these cases. There were no differences at the level of significance after excluding these cases from the previous results.

  • Clear information is provided on age, sex, and family history whereas no information is provided on medical treatment before or in context of ITP. Also no statement is given whether which line of treatment was given.

Response: Thank you for your insightful comments. In fact, we recruited cases and collected samples at the time of their initial presentation, prior to the start of any treatment, to rule out any potential impact of therapy on AIRE levels. Furthermore, we only had five cases with platelet counts below 30×103/μl that required therapy (four of them received first-line corticosteroid therapy, and one case received TPO agonist), and we also collected the samples prior to administering any treatment to them. We clarified the sampling timing in the Sampling section (4.3, Line: 260).

  • If treatment had been applied prior to inclusion in the study could that have an impact on the AIRE serum levels due to immune transformation?

Response: Thank you for your insightful comments and we totally agree with your opinion. Since corticosteroids are the primary treatment for ITP, their immunosuppressive effects would inevitably have an effect on AIRE serum levels. However, no medical treatments were administered to our cases prior to their inclusion in the study, which rules out any potential effect of therapy on AIRE levels. However, we have added this important point to our limitations and we also recommend that our upcoming studies will investigate the effect of therapy on AIRE serum levels (Lines: 229-234).

  • The IWG classifies ITP on duration of platelet count <100 x 109/L (Newly diagnosed             <3-mo duration, Persistent                 3-12-mo duration, Chronic            >12-mo duration) as duration has proven to show a great impact on outcome. Underlying causation is most likely attached to progression of immune disorder. Information should be provided on duration and AIRE level. Is there a correlation?

Response: Thank you for your valuable comment. In fact, we collected samples for estimating AIRE levels at the time of diagnosis, so follow-up on some patients had not yet been completed in order to appropriately classify them. We were therefore unable to establish a correlation between AIRE level and duration of illness. However, we have added this important point to our limitations and we also recommend that future research be conducted on the correlation between AIRE level and ITP duration (Lines: 231-234).

Minor:

  • As the authors point out emphasis has been put only SNPs rs2075876 and rs760426 as they are linked particularly to RA and SLE which are both not primarily linked to ITP. What has been causative for prioritising the chosen SNPs?

Response: Thank you for your insightful comment. As we stated in the discussion section, we chose these two polymorphisms because, of the eleven SNPs of AIRE, only these two SNPs, rs2075876 and rs760426, received attention in previous studies and were found to be risk factors for a variety of autoimmune disorders, including RA and SLE. RA and SLE share some pathogenicity with ITP in terms of altered immune tolerance. Although the mechanisms underlying RA and SLE progression are unknown, evidence suggests that loss of self-tolerance with subsequent activation of the autoreactive T and B cells, autoantibody production, and eventual inflammation-induced tissue injury and organ dysfunction, plays a role (1, 2). In the same vein, ITP is characterized by the excessive activation and proliferation of platelet auto-antigen reactive cytotoxic T lymphocytes (3), abnormal T helper cells (4, 5), and B cells which produce anti-GPIIb/IIIa antibodies that destroy platelets (6). Furthermore, these two SNPs have been found to affect AIRE protein coding and thus its expression, altering immune tolerance. As a result, we assumed that these could be linked to ITP, given that it is also an autoimmune and inflammatory disease. We explained this rationale in the discussion section (Lines: 152-162). In addition, we included a suggestion for future research into other AIRE SNPs in ITP (Line: 232).

References:

  • 1- Mellado, M. et al. T Cell Migration in Rheumatoid Arthritis. Front. Immunol. 6, 1–12 (2015).
  • 2- P. D’Cruz, M. A. Khamashta, and G. R. Hughes, “Systemic lupus erythematosus,” Lancet, vol. 369, no. 9561, pp. 587–596, 2007.
  • 3- Zhang XL, Peng J, Sun JZ, Guo CS, Yu Y, Wang ZG, Chu XX, Hou M. Modulation of immune response with cytotoxic T-lymphocyte-associated antigen 4 immunoglobulin-induced anergic T cells in chronic idiopathic thrombocytopenic purpura. J Thromb Haemost. 2008;6:158–165. doi: 10.1111/j.1538-7836.2007.02804.x.
  • 4- Wang T, Zhao H, Ren H, Guo J, Xu M, Yang R, Han ZC. Type 1 and type 2 T-cell profiles in idiopathic thrombocytopenic purpura. Haematologica. 2005;90:914–923.
  • 5- Shan NN, Ji XB, Wang X, Li Y, Liu X, Zhu XJ, Hou M. In vitro recovery of Th1/Th2 balance in PBMCs from patients with immune thrombocytopenia through the actions of IL-18BPa/Fc. Thromb Res. 2011;128:e119–e124.
  • 6- Kuwana, M., Okazaki, Y., Ikeda, Y., 2014. Detection of circulating B cells producing anti-GPIb autoantibodies in patients with immune thrombocytopenia. PLoS One 9 (1), e86943.

  • As platelet counts > 30 x 109/L show no clinical implications on higher rate of bleeding/ severe bleeding and therefore remain commonly untreated it would have been of significance including patients presenting with < 30 x 109/L.

Response: Thank you for your professional comment and we totally agree with your opinion. We included all cases of primary ITP, regardless their platelet count, in our study. Only five cases in our study had platelet counts of less than 30×103/μl, and were indicated for therapy. Unfortunately, they are insufficient to be statistically analyzed as a subgroup.

==================================================================================

Finally, we would like to express our gratitude to the editors and reviewers for their time, efforts and their valuable comments which helped us to improve this manuscript.

Best regards;

Muhammad Abdel Ghafar

Reviewer 2 Report

The authors present a well written communication that evaluated the influence of autoimmune regulator (AIRE) and the risk of ITP in a Egyptian population. SNPs were evaluated in rs760426 A/G and rs2075876, since previous evidence showed an association between polymorphisms and the occurrence of rheumatoid arthritis and systemic lupus erythematosus. rs2075876 AA genotype and A allele were associated with an increased risk of ITP in contrast to rs760426, which shows no correlation. The occurrence of these genotypes was also associated with serum AIRE levels establishing a plausible causality.

Even though other polymorphisms involved with auto-tolerance have been suggested to play a role in ITP, this is the first publication to show an association between AIRE polymorphisms and the disease, therefore containing novel information in this field.

This paper is in general well written, and the conclusions are supported by the methodology. There are some small concerns that could be addressed to improve the paper. 

- ITP is normally an exclusion diagnosis. What were the criteria used to diagnose the disease in this population?

- SLE is a complex autoimmune disease that can also cause thrombocytopenia and it is an important differential diagnosis to ITP. As the authors report, SNPs in both genes have been reported to be associated with SLE. Therfore, SLE is an important confounder in this study. Did the authors exclude patients with SLE? 

Furthermore, were patients with rheumatoid arthritis also excluded from the study (both in the ITP and Control group?).

Please report the inclusion and exclusion criteria, so that the populations are clear.

In the discussion it would be interesting to report some other SNPs involved in immune self-tolerance that have been shown to play a role in ITP. 

Please discuss the effect size of the correlations of between the relevant polymorphisms and how does this compare to other SNPs in the literature. 

Author Response

Dear editor /referee,

Many thanks for your constructive and valuable criticisms. Our responses are presented below and we are looking forward and ready to respond to any future comment (s)

Reviewers' comments:

Reviewer #2:

The authors present a well written communication that evaluated the influence of autoimmune regulator (AIRE) and the risk of ITP in the Egyptian population. SNPs were evaluated in rs760426 A/G and rs2075876, since previous evidence showed an association between polymorphisms and the occurrence of rheumatoid arthritis and systemic lupus erythematosus. rs2075876 AA genotype and A allele were associated with an increased risk of ITP in contrast to rs760426, which shows no correlation. The occurrence of these genotypes was also associated with serum AIRE levels establishing a plausible causality.

Even though other polymorphisms involved with auto-tolerance have been suggested to play a role in ITP, this is the first publication to show an association between AIRE polymorphisms and the disease, therefore containing novel information in this field.

This paper is in general well written, and the conclusions are supported by the methodology. There are some small concerns that could be addressed to improve the paper. 

Response: Thank you very much for recommending our manuscript and your encouraging comments and we are pleased to respond to them.

- ITP is normally an exclusion diagnosis. What were the criteria used to diagnose the disease in this population?

Response: Thank you for your professional comment. Our cases were diagnosed as primary ITP based on the International Working Group criteria as isolated thrombocytopenia <100×103/μl in the absence of other causes or disorders associated with thrombocytopenia. This was clarified in the 4.1. Study cohort section (Lines: 239-242).

- SLE is a complex autoimmune disease that can also cause thrombocytopenia and it is an important differential diagnosis to ITP. As the authors report, SNPs in both genes have been reported to be associated with SLE. Therefore, SLE is an important confounder in this study. Did the authors exclude patients with SLE? 

Furthermore, were patients with rheumatoid arthritis also excluded from the study (both in the ITP and Control group?).

Please report the inclusion and exclusion criteria, so that the populations are clear.

Response: Thank you for your insightful comment. As stated in the original manuscript, patients with secondary causes of ITP, such as RA and SLE, were excluded from the study in both ITP and control subjects. We clarified the inclusion and exclusion criteria based on your professional advice as follows (Lines: 237-247):

“In this case-control study, 96 patients with primary ITP at their initial presentation were recruited from the hematology unit of the Internal Medicine Department, Tanta University Hospitals as a case group. Their inclusion criteria were all cases diagnosed with ITP based on the standard criteria of the ITP working group as isolated thrombocytopenia (platelet counts of less than 100x103/µL in the absence of other causes of thrombocytopenia [36]. Patients with secondary causes of ITP such as viral infections, drugs, and autoimmune disorders such as RA, SLE, Chron’s disease were excluded from the study.

In addition, 100 healthy subjects with normal platelet counts were recruited from those attending the out-patient clinics at Tanta University Hospitals as a control group. The exclusion criteria of case group were also applied to the control group.“

In the discussion it would be interesting to report some other SNPs involved in immune self-tolerance that have been shown to play a role in ITP. 

Response: Thank you for your professional comment. In the discussion section, we included a paragraph demonstrating other SNPs involved in immune tolerance that had been studied in relation to ITP based on your professional advice (Lines: 139-144).

Please discuss the effect size of the correlations of between the relevant polymorphisms and how does this compare to other SNPs in the literature. 

Response: Thank you for your insightful comment. Based on your professional recommendations, we discussed the effect size of our two studied SNPs as odds ratio for the A-A haplotype (OR: 1.821 95% CI: 1.099-3.017) and present other risk ratios for other AIRE SNP haplotypes in other autoimmune diseases in the discussion section (Lines: 197-207). We also analyze the variant effect predictor using this online tool (http://grch37.ensembl.org/Homo sapiens/Tools/VEP) and present the possibility of genetic location for our two studied AIRE SNPs as well as other studied AIRE SNPs in the discussion section (Lines: 207-211).

==================================================================================

Finally, we would like to express our gratitude to the editors and reviewers for their time, efforts and their valuable comments which helped us to improve this manuscript.

Best regards;

Muhammad Abdel Ghafar
